# Occurrence and Characteristics of Cut Injuries in Hedgehogs in Germany: A Collection of Individual Cases

**DOI:** 10.3390/ani14010057

**Published:** 2023-12-22

**Authors:** Anne Berger

**Affiliations:** Leibniz Institute for Zoo and Wildlife Research, Alfred-Kowalke-Straße 17, 10315 Berlin, Germany; berger@izw-berlin.de

**Keywords:** European hedgehog, *Erinaceus europaeus*, rehabilitation centre, cut injuries, mortality rate, wildlife conservation, human–wildlife conflict, animal suffer, animal welfare

## Abstract

**Simple Summary:**

The European hedgehog is a protected species, but its populations are declining across Europe. This decline has various causes, such as lack of food, habitat loss and fragmentation or life-threatening injuries caused by human activities. Hedgehog rescue centres increasingly report hedgehogs found with severe cuts, presumably caused by garden tools. Responsibility for hedgehog injuries caused by robotic lawnmowers and possible technical or political solutions to prevent such injuries are currently being discussed between wildlife conservationists, mower manufacturers and politicians. This discussion has so far lacked basic data on the extent of cutting injuries in hedgehogs. In this study, data on hedgehogs with cut injuries were collected throughout Germany in order to gain an impression of where, when and how frequently these injuries occur. The number of reporting hedgehog care centres and thus the number of hedgehogs reported per federal state varied highly. Out of the total of 370 injured hedgehogs reported, at least 60% were found over 12 h after the accident and at least 47% did not survive as a result of the injury. Overall, this study shows that cutting injuries caused by garden maintenance equipment pose an additional lethal danger to this declining, protected wildlife species.

**Abstract:**

The number of European hedgehogs (*Erinaceus europaeus*) is in long-term decline across Europe. Recently, an additional threat to hedgehogs’ lives has been cutting injuries caused by garden care equipment, but to date, there have been no reliable data on their spatial and temporal occurrence as well as characteristics such as mortality rate. Usually, found injured hedgehogs are admitted to care centres. In this study, data on hedgehogs with cutting injuries were collected from care centres throughout Germany. Over a period of 16 months, data on a total of 370 hedgehogs with cut injuries were reported by 71 care centres. At least 60% of these hedgehogs were found more than 12 h after the accident and at least 47% did not survive as a result of the injury. The comparatively high mortality rate coupled with a possible high number of unreported cases of hedgehogs with laceration injuries show that these accidents pose an additional, serious danger to hedgehogs, both impacting the welfare of individual animals and having a broader effect on the conservation potential of this species. Moreover, the data collected objectify the current discussion on the need for possible technical or political solutions to prevent such injuries.

## 1. Introduction

The West European hedgehog (*Erinaceus europaeus*, hereafter referred to as “hedgehog”) is a solitary, nocturnal insectivore and is one of the most popular and well-known wild animals among the general population [1]. People’s interest in these animals is probably due to the fact that they are often found in close proximity to humans. Human habitations, especially gardens, have structures that are particularly attractive to hedgehogs, such as a high proportion of green areas, bushes or additional sources of water and food [2]. Despite their high popularity, hedgehog populations have experienced a serious and continuous decline in recent decades [3,4,5,6,7], especially in rural areas [8,9]. Reasons for this decline include habitat loss and fragmentation [9,10,11], reduced food availability, partly due to the use of pesticides and insecticides [12,13], in some areas, also intra-guild predation by badgers [8,14,15,16] and possible climate-related effects [17]. Moreover, there are often various fatal accident hazards to which hedgehogs are exposed, such as road traffic or entanglement in garbage [18,19,20].

Hedgehogs are subject to protection in large parts of Europe; in Germany, they are specially protected wild animals that cannot be hunted and may not be caught, injured or killed [21]. However, the law does grant one exception to the ban on possession: it states that injured, helpless and/or sick animals may be taken into human care in order to nurse them back to health. As soon as the hedgehogs are able to maintain themselves independently, they must be released into the wild immediately. Caring for sick, injured or orphaned wild animals and their preparation for release back into the wild are usually carried out in wildlife rehabilitation centres. The rehabilitation of wild animals requires large investments of time, personnel and money [22] and therefore it is often important to ask which animals can be taken in at all, that is, if they have a high chance of surviving well during care and also after release [23]. Hedgehogs are among the European wildlife species that are most frequently rehabilitated by humans [24], often not even in public rehabilitation centres, but on a private, voluntary, non-profit basis.

Many of these hedgehog care centres report a significant increase in the incidence of cutting injuries caused by garden maintenance equipment (scythes, string trimmers, robotic lawnmowers). As the global market for robotic lawnmowers is expanding at an annual growth rate of more than 12% in the period 2019 to 2025 [25], and as the use of robotic mowers is increasing significantly compared to other maintenance equipment, it could be hypothesised that the increase in the number of cut injuries in hedgehogs is associated with the use of these robotic mowers. An initial study has already shown that—contrary to the manufacturer’s specifications—many models of robotic lawnmowers can cause serious cutting injuries to hedgehogs [26]. A great attractiveness of robotic lawnmowers stems from the fact that, unlike other lawnmowers, these devices can be used legally for an unlimited period of time (i.e., also at night and on public holidays and on Sundays) due to their low noise emissions. They can also work unattended (i.e., in the absence of humans). These two characteristics in particular make it very likely that many collisions between robotic lawnmowers and nocturnal hedgehogs occur and that these are often not even noticed by humans, especially in cases where the hedgehog is only slightly injured and still able to run away from the scene after the collision.

In Germany, there have already been many petitions and political efforts by hedgehog protection and care organisations calling for a general ban on the night-time use of mowing robots [27,28,29]. So far, these have all failed due to a lack of public interest, a lack of data on the extent of cut injuries in hedgehogs or a lack of legislation to enforce these rules at the regional level. The aim of this study is to collect and quantify concrete data on the temporal and spatial distribution of these injuries, specific characteristics of the wounds, the probability of survival and the extent of the care required for hedgehogs found with cut injuries in Germany. These figures are intended to add an objective perspective to the emotionally charged debate on hedgehogs with cut wounds and are fundamental to the current discussion on the need for possible technical or political solutions to prevent such injuries.

By statistically analysing the individual cases collected, the following questions were investigated:

(A) Are there days of the week when there are significantly more cases of cut injuries in hedgehogs or when injured hedgehogs are found more frequently? On Sundays and public holidays, and generally between 8 p.m. and 6 a.m. (at night, during the natural activity period of hedgehogs), the use of garden maintenance devices is prohibited by law for noise protection reasons and no maintenance work is carried out by public green maintenance authorities at weekends (Saturday and Sunday). An exception to this is robotic lawnmowers, which are not subject to these time restrictions, as they are usually quieter than the noise limits to be observed [30] and are therefore also used on Sundays and at night and are preferable in private areas. An above-average incidence of hedgehogs injured by cutting on Sundays would suggest that they are predominantly injured by robotic lawnmowers, especially those for private use;

(B) How are the cases of hedgehogs found with cutting injuries distributed across different age and sex groups?

(C) Are there individual characteristics of the injuries that lead to an increased mortality risk?

## 2. Materials and Methods

### 2.1. Data Collection

All data on cases of hedgehogs with cut injuries in this study were collected via a Facebook group set up on 28 June 2022. Due to graphic nature of the photos of reported animals and to protect personal data, the Facebook group was not accessible to the general population but individually to people who had come into direct contact with cut hedgehogs and could provide information about them. Every report of a cut hedgehog contained the following *minimal information*:Date on which the animal was found;Place where the animal was found;One or more photos showing the injuries;Who treated or diagnosed the animal or who provided initial treatment and care for the injured hedgehog (this was asked to ensure that *additional* diagnostic *information* was provided by expert veterinarians or experienced hedgehog carers).

The following *additional information* was provided, where known:Sex of the animal;Estimated age of the animal;Fate of the animal (euthanasia/died while in care/recovered/released back into the wild).

All information from the reports that contained at least the *minimal information* was transferred to a table; thereby, information on the wound characteristics was taken from the photos or diagnostic descriptions sent in. The table contained the following columns with the following (categorical) content entries:Hedgehog identification number: xxx;Date (day of found): YYYYMMDD;Day of the week: Monday/Tuesday/Wednesday/Thursday/Friday/Saturday/Sunday;Location: exact address or at least the postcode;Sex: male/female/reproducing female (means pregnant, lactating)/unknown;Age: adult (from the survival of the first hibernation, note: all animals found from January to May were always considered adult here)/juvenile (heavier than 100 g and before the first hibernation)/nestling (less than 100 g)/unknown (age was not reported or cannot be estimated from the reported information);Fate: euthanasia/died (during treatment or rehabilitation)/survived or released into the wild/unknown (this includes cases where no information was provided, but also all cases that are still open with regard to survival, e.g., still undergoing treatment);Characteristics of the wound:Size of the cut surface larger than 2 × 2 cm: yes/no/unknown (all cases were listed as “unknown” for which the photos did not allow a clear size estimation, as reference size objects—such as the fingers or hands of the treating person—were missing);Presence of maggots: yes/no/unknown;Presence of necrosis: yes/no/unknown;Presence of abscesses: yes/no/unknown;Bone damage (e.g., fractures, splintering): yes/no/unknown;Loss of body parts: yes/no/unknown;Age of the wound: <12 h (the injured animal was noticed during the accident and taken to the vet or no maggots or necrosis were recognisable on the wound or mentioned in the reports)/>12 h/already healed (cut wound is already scarred and overgrown)/unknown (all cases were classified here in which an assessment could not be clearly made on the basis of the photos and reports). This wound age estimate could be made “remotely” from photos and reports based on the knowledge that fly maggots need 8–12 h to hatch from the fly eggs even under ideal conditions [31,32,33], thus wounds with maggots had to be at least 8–12 h old. According to textbooks on wound healing processes in wild animals [34,35] and to personal reports on the duration of successful wound healing in hedgehogs from care stations, the age of wounds that have already healed can be estimated at around 1 week up to several months.

Body parts affected:Head (in front of the imaginary line between the ears): yes/no/unknown;Neck and shoulders (starting behind the imaginary line between the ears): yes/no/unknown;Back (spines on the back up to the edge): yes/no/unknown;Extremities: yes/no/unknown;Flank/belly (from the edge of the spines towards the belly): yes/no/unknown.

### 2.2. Statistical Analysis

Analyses were performed using the basic package R Studio [36]. Pearson’s chi-square tests were used to check whether (1) all animals were found and (b) all animals with a wound age <12 h were found at the expected frequency on each day of the week (*p* = 1/7).

For the following data analysis, the category “fate” was pooled into two definitions: “died” (all animals that were euthanised or died during care) and “survived” (all animals that survived). All cases where the fate of the animals was not known (either not reported or the animals are still under treatment) were excluded from the following calculations.

For the following parameters (A) wound size, (B) maggots, (C) necrosis, (D) injured bones, (E) abscess, (F) severed body parts, (G) head, (H) neck, (I) back, (J) extremities, (K) flank/abdomen and (L) wound age, the number of “died” (g) versus “survived” (s) animals was counted and the mortality rate (mr = g × 100/(g + s)) was calculated.

To investigate whether the age of the wound, the presence of certain wound characteristics (A–F) or the affected body part (G–K) influenced the fate of the animal “died”/”survived”, the statistical significance for each of these frequency ratios was tested using chi-square tests.

## 3. Results

The Facebook group was set up on 28 June 2022. By 31 October 2023 (during 16 months), a total of 370 cases of hedgehogs with cut injuries were reported by 71 reporters (average: 5.2, median: 2.0). Figure 1 shows the locations where the hedgehogs were found on the map of Germany, separated by colour and by symbol according to the information on their fate (euthanised or died during care versus survived or still in care or fate is unknown). Figure 1 also shows the 17 hedgehog care centres that reported an above-average number (n > 5) of hedgehogs. The majority of the 71 reporters were small hedgehog care centres with only a few hedgehogs in their care; thus, more than half reported only one hedgehog (n = 27) or two hedgehogs (n = 14). Table 1 shows the number of hedgehogs found and the number of hedgehog care centres that reported more than five hedgehogs in each German federal state.

The earliest find data are from 2013; Figure 2 shows how many hedgehogs were found per year and Figure 3 shows the find data per month, also showing the proportion of animals found in 2023, in 2022 and in previous years.

Although the distribution of hedgehogs found by day of the week shows that fewer animals were found on Fridays than on other days and that a particularly large number were found at weekends and on Tuesdays, these differences are not significant (chi-square test for given probabilities for hedgehogs found within 12 h of the accident: x-squared = 7.6333, df = 6, *p*-value = 0.2662; for hedgehogs which were found later than 12 h after the accident: x-squared = 2.3457, df = 6, *p*-value = 0.8853) (Figure 4).

Out of the 370 reported hedgehogs, 115 were euthanised due to the severity of the injury, 60 died during treatment or further care, 120 survived the treatment or were released back into the wild and there was no information on 75 or their care was still ongoing (Figure 5). A total of 32.7% (n = 121) of the hedgehogs were found within the first 12 h after the injury, 44.9 % (n = 166) of the hedgehogs had wounds older than 12 h and 14.6% (n = 54) had ones older than several weeks. A total of 33.2% (n = 123) of the hedgehogs were males, 28.4% (n = 105) were females (of which 3.5% (n = 13) were currently pregnant or lactating); the sex of 142 animals (38.4%) was unknown. Among the reported hedgehogs, 5 (1.4%) were dependent nestlings (<100 g), 35 (9.5%) were identified as juveniles and 191 (51.6%) were classified as adults (Figure 5).

Figure 6 shows the number of hedgehogs that “died” and “did not die” and the resulting mortality rate calculated for the respective wound characteristics or wounded body parts. If the mortality rate is >50%, this means that more than half of the animals exhibiting this wound characteristic or wounded body part died. The results of the chi-square test, which tested whether the certain wound characteristics or wounded body parts had an effect on the fate of the animal (died or survived), are shown as results with asterisks in Figure 6 and as specific *p*-values in Table 2.

## 4. Discussion

In this study, data on hedgehogs found with cut injuries throughout Germany were compiled for the first time and statistically analysed for their spatial and temporal occurrence and injury characteristics.

Although the spatial distribution of the hedgehogs found suggests that some parts of Germany (particularly North Rhine-Westphalia and Lower Saxony) have a particularly high incidence of cut injuries in hedgehogs (Figure 1), it should be considered that most of the larger hedgehog rescue centres that took part in the collection are located in these areas of high occurrence. When asking at other larger hedgehog centres in Germany that did not take part in the data collection of this study, we were told that they also have many hedgehogs with cuts in their care, but that they do not have the capacity to document or report this information as they are too busy caring for the hedgehogs, whose welfare comes before documentation. Moreover, most German hedgehog centres use an analogue protocol system, which means that queries about certain types of wounds require considerable effort for them. Apart from the fact that there is no complete and, above all, up-to-date list of all hedgehog care centres operating in Germany, many of the well-known and, above all, larger hedgehog centres did not participate in the collection of the data presented here, despite being asked. Therefore, the spatial distribution of the 370 hedgehogs found in this study only reflects a section of the overall German situation; areas without reports do not mean that there are no hedgehogs injured by cuts there.

Nevertheless, the map also shows that there is a very large number of hedgehogs with cut injuries, particularly around the larger hedgehog centres. Even if the area from which hedgehogs in need of help are brought to the (especially larger) hedgehog centres is very large [37], the willingness of people to take an injured hedgehog to a vet or care centre located far away is certainly limited at some point. Therefore, hedgehogs can of course only be helped where there are well-known and specialised hedgehog care centres or vets nearby. From the reports on the hedgehogs that could be released back into the wild after treatment (n = 120), it was clear that cut hedgehogs usually take many months to recover, and the treatment descriptions also suggest that several or complicated surgeries and expensive medication might be necessary. A study of 11,801 hedgehog patients from seven hedgehog centres in Germany showed a duration of stay of 0–359 days (mean 65.9 days, spread 77.2 days) [37]; hedgehogs with cut injuries therefore require above-average and likely more expensive treatment than other hedgehogs in need of care. Many centres also reported that the steadily increasing number and, above all, the severity of the injuries were pushing them to their spatial, financial, physical and mental capacity limits and that they would either have to close or would no longer be able to take in any more hedgehogs if this trend of rising number of hedgehogs with cut injuries continued.

The data in our study (Figure 2 and Figure 3) show that cut injuries in hedgehogs are becoming increasingly common. Though the collection of data only started on 28 June 2022 and continued for 16 months, the 4 months (July–October) in which data were collected and reported for both 2022 and 2023 show that there was an increase in the number of cases from 2022 to 2023 (as in all other months too). The distribution of the findings by month (Figure 3) shows that in 8.6% of cases, cuts also occurred during the hedgehog hibernation period (January to March), a time when the animals generally do not leave their hibernation nest [17]. Studies on the time of admission of hedgehogs to rehabilitation centres show that hedgehogs are brought in throughout the year with a bimodal distribution pattern, with a peak in summer with early litters (July–August) and one larger peak in autumn (August to November) with late litters and animals that are too weak to hibernate [24,38,39]. Most of the cut hedgehogs in this study were reported in May and June (Figure 3), a time when there have always been very few new hedgehogs in need of care at hedgehog centres [38,39,40,41]. As the majority of hedgehogs with cut injuries are also long-term patients, these high numbers in early summer (i.e., before the previous peak periods) mean that important inpatient capacities (e.g., hedgehog boxes) are already occupied resulting in no space for the many animals that come in summer or autumn. The high number of hedgehogs injured by cuts from May to July is due to both hedgehogs’ way of life (hedgehogs are active and males travel longer distances and explore unknown terrain during the mating season [17]), but also to human activities in garden and green space maintenance. Even though this study shows a trend that a particularly high number of hedgehogs with fresh cuts were found on Tuesdays, Saturdays and Sundays and particularly few on Fridays (Figure 4), which gives an indication of the day of the week when the hedgehogs were injured by garden maintenance works, these differences are not significant and may therefore have arisen by chance. However, since robotic lawnmowers are the only gardening tools that can legally be used at any time, including Sundays and at night, the results also suggest that many of the injuries could have resulted from collisions with robotic lawnmowers; since the use of all other devices is not allowed on Sundays for noise protection reasons [30], there would also be fewer cuts on Sundays if these devices were mainly responsible for these cut injuries. Hedgehogs with older cuts were found frequently (although not statistically significantly) on Saturdays, Sundays and Mondays and rarely on Fridays, which gives an indication of when people are in green spaces/gardens and become aware of injured hedgehogs.

This study showed a mortality rate of at least 47.3% (in 20.3% of cases, the fate of the animals is unknown) (Figure 5). Long-term studies in other European rehabilitation centres report a mortality rate of approximately 1/3 of the total number of animals admitted [24,42], but it is apparent that the mortality rate is highly dependent on the reason for admission, the time of year (which in turn is related to the reason for admission) and the respective care centre (some centres specialise in intensive care patients with an increased probability of mortality) [24,37,38,39,42,43]. In addition, data from many rehabilitation centres show a decreasing mortality rate over the years, probably due to better treatment options and care practices [43]. In this respect, although the number of animals that died and, in particular, were euthanised in our study is high, it is comparable with data from other studies.

In our study, 32.7% of the animals were found within 12 h of the cut injury (Figure 5). The majority of animals were not found until several days or even weeks after the accident. These animals were often no longer able to search for food or eat on their own or to lick their wounds, so that their wounds became infected or flies laid their eggs there. Animals with such wounds have little chance of surviving in the wild. In 14.6% of the animals in our study, the wounds had nearly healed and some even appeared to be able to continue living with the injury, but here too there were individual cases with only a very low probability of survival (they were severely dehydrated, emaciated or missing several limbs). In any case, these injured animals suffer prolonged severe pain, suffering and harm which are caused by human action and which according to European and German animal welfare law can only be excused with a reasonable cause and must be avoided or limited as far as possible [44].

Out of the 370 animals reported, 33.2% were male and 28.4% female (Figure 5); in other studies, the sex ratio was approximately equal [37,38,42,45] or the injured males outnumbered the females due to their wider ranging behaviour and the resulting higher accident risk [43]. In this study, 3.5% of the reported hedgehogs were pregnant or lactating females at the time of the accident, thereby lowering the survival possibility their young, too. The vast majority of animals in this study were adults (Figure 5), but there were also dependent nestlings with lacerations which had sustained these injuries through the destruction of the litter nest and not through their own movement behaviour. The relatively high number of adult animals compared to other studies on hedgehog care centres [38,39,42,45] can be explained by the fact that subadult animals were also assessed as adults in this study, as a more precise age estimate is in general quite difficult and was not possible based on the reported information [46,47,48]. In this study, the proportion of animals for which the sex or age category was unknown is quite high, as this information was mostly not provided by the reporters as it was not demanded as necessary (minimal) information.

Even though the criteria for wound assessment in this study were chosen to be quite simple so that they could be made remotely based on photos and diagnostic reports, it must be mentioned that this methodology can be very error-prone since the diagnostic reports and photos are from many different veterinarians and hedgehog keepers, whose assessments and working methods can sometimes differ greatly from one another. A further limitation of the “remote” methodology when assessing wound characteristics or the sex or age of the animal is shown by the high proportion of “unknown” case assignments; in order to avoid incorrect assignments, the rating “unknown” was often given, which can falsify overall statistics or make them difficult to interpret.

This study showed that the animals that died (47.3%) were significantly more likely to have cuts to the head, flank and abdomen (Figure 6). There were also significant correlations between the occurrence of abscesses and removed body parts. The body parts that were cut off were noses, eyes, ears, snouts, toes, feet or whole legs. With these kinds of wounds, compared to other types of cut injuries (e.g., in the neck or very large wounds), there is virtually no chance of healing and survival. Other parameters, such as the age of the wound, had no influence on the probability of survival of the injured animal, which means an average probability of survival even if the injured animal was found days to weeks after the accident. However, due to their small body size, their hidden, nocturnal way of life and their danger-avoiding behaviour, it is rather coincidental that injured hedgehogs are found and the number of unknown cases of injured animals that are not found must be considered high. Hedgehogs try to behave as inconspicuously as possible when injured or try to find shelter in bushes in order to avoid attracting the attention of potential predators such as crows or foxes [49]. This behaviour also explains the high number of hedgehogs that were only found days or weeks after the injury. However, some individuals in this study, particularly those with extreme head wounds, were only found because they sought out human’s vicinity on their own. But even hedgehogs that have died from cuts in the wild are not that easy to find: either they reached shelter before their death and are difficult to discover there, or other wild animals attacked and fed on them, and their carcasses disappeared relative quickly [50].

Various studies have examined the causes of death in hedgehogs in the wild and their possible impact on the population [20,51,52]; however, estimations of impact on the population always need solid figures about the population itself. Thanks to many years of citizen science monitoring, there is already a relatively solid database on the hedgehog population in Great Britain [7], which, for instance, made it possible to estimate the yearly number of hedgehogs which become life-threateningly entangled in garbage and the influence of this number on the development of the hedgehog population [20]. However, such figures on the number of hedgehogs in Germany are missing; thus, such estimates cannot be made with the data from this study. This will therefore be the subject of further investigations, because politicians and society first demand reliable information on the extent of hedgehogs with cut injuries and the influence of this phenomenon on their population development before they introduce restrictions (such as a ban on night-time use of robotic lawnmowers). Nevertheless, this study was able to prove that hedgehogs with cuts caused by humans are not rare, isolated cases, but rather represent a problem that is relevant to animal welfare and requires technical (like devices that can be programmed for a specific time of day and thus can only be used in daylight hours) or political solutions as soon as possible.

## 5. Conclusions

With the help of several hedgehog care centres, data on hedgehogs injured by cuts across Germany were compiled. The analysis of these data showed that cut injuries increase from year to year, placing an enormous burden on many hedgehog care centres and using up important resources, as these injuries often require above-average care and treatment. There is also a considerable animal welfare problem, as the majority of hedgehogs with cut injuries are found days or weeks after the accident and therefore have to endure considerable suffering, pain and harm over a long period of time. Such animal suffering is prohibited by law, provided there are alternatives that do not cause animal suffering. At the very least, alternatives that do not cause that much animal suffering are certainly available through the technical or political implementation of a ban on night-time use of robotic mowers and these must be implemented immediately, which this study has attempted to contribute to.

## Figures and Tables

**Figure 1 animals-14-00057-f001:**
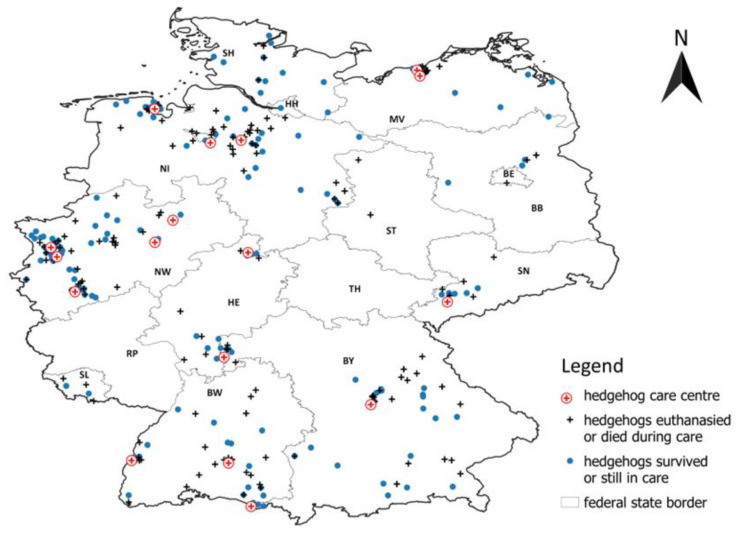
Map of the locations in Germany in which the 370 reported hedgehogs with cut injuries were found, divided according to their fate (black cross = did not survive, blue dot = did survive or fate is unknown), and of 17 hedgehog care centres that reported more than 5 of these hedgehogs. Uppercase letters give the German federal state abbreviations, see Table 1.

**Figure 2 animals-14-00057-f002:**
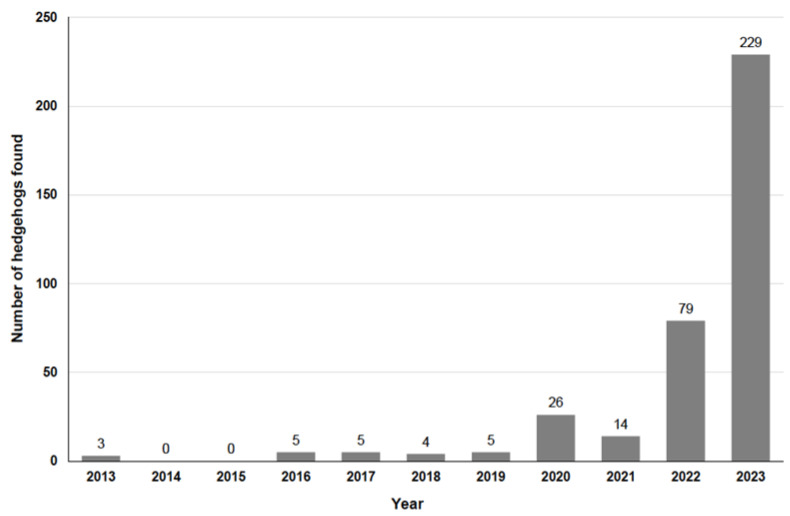
Number of 370 reported hedgehogs plotted according to the year in which they were found.

**Figure 3 animals-14-00057-f003:**
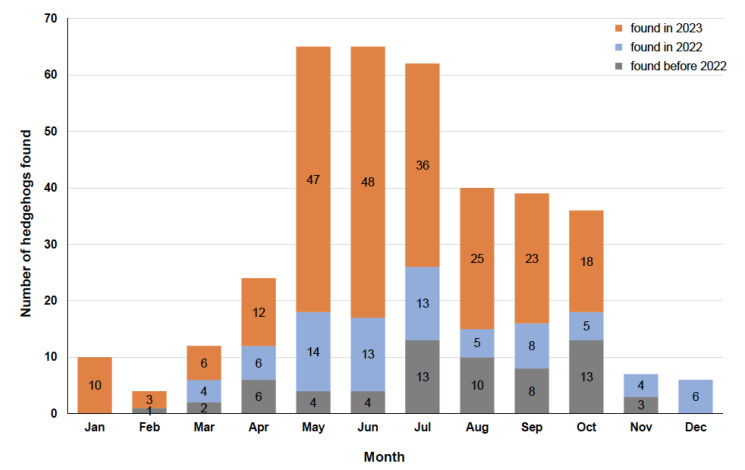
Number of 370 reported hedgehogs plotted according to the month in which they were found. The respective proportions of finds from 2023 (orange), 2022 (blue-grey) and the years before 2022 (grey) are colour-coded.

**Figure 4 animals-14-00057-f004:**
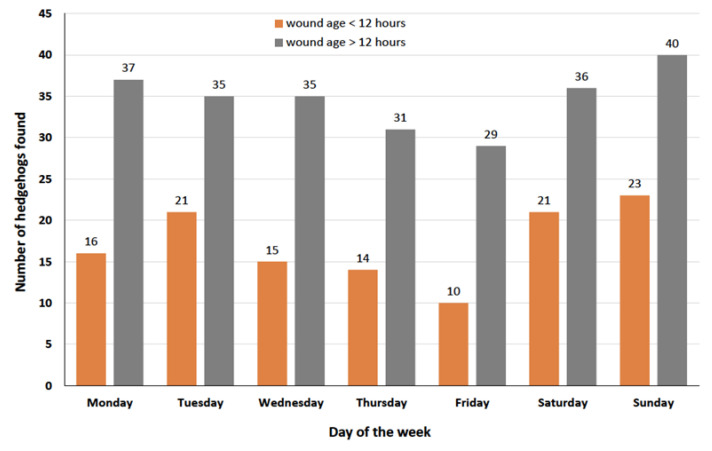
Number of 363 injured hedgehogs plotted according to the day of the week on which they were found: orange: hedgehogs found within 12 h of the accident (x-squared = 7.6333, df = 6, *p*-value = 0.2662), grey: hedgehogs whose wounds were older than 12 h on the date they were found (x-squared = 2.3457, df = 6, *p*-value = 0.8853). In 7 cases, the exact date and therefore the day of the week on which the injured hedgehog was found was not known.

**Figure 5 animals-14-00057-f005:**
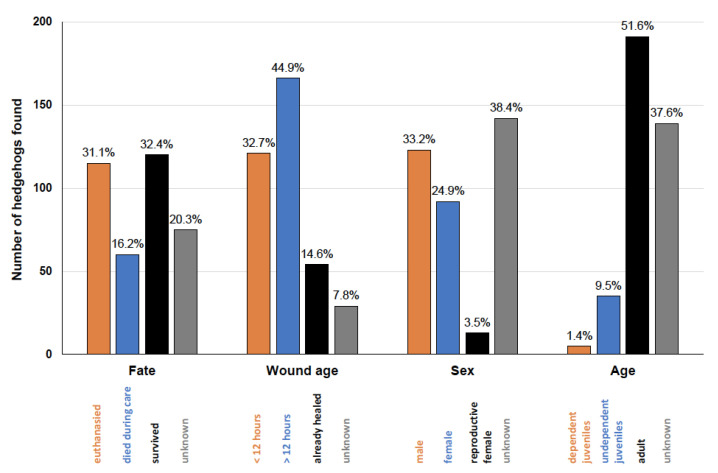
Number and percentages of the different fates (euthanised/died during care/survived or released/unknown), age of the wound when the hedgehog was found (<12 h/>12 h/already healed/unknown), sex (male, female, reproducing female/unknown) and age of the hedgehog (dependent nestling/independent juvenile/adult/unknown) of the 370 reported hedgehogs found with cut wounds.

**Figure 6 animals-14-00057-f006:**
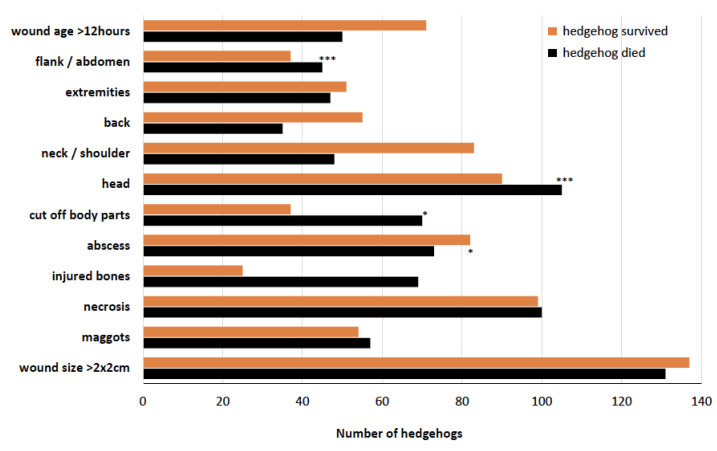
Number of hedgehogs with certain wound characteristics (wound is larger than 2 × 2 cm/presence of maggots/necrosis/bone injuries/body parts cut off/wound is older than 12 h) or whose wounds occurred on certain parts of the body (head/neck or shoulders/back/extremities/flank or abdomen) (multiple counts possible). The numbers are divided into hedgehogs that died by euthanasia or during treatment/care (black) and hedgehogs that survived. Cases where the fate of the animals was not known are not included. Significance results for the ratio of the parameters to the overall mortality of the 370 reported hedgehogs are shown symbolically (* *p*-value < 0.05, *** *p*-value < 0.001).

**Table 1 animals-14-00057-t001:** Overview of the number of reported hedgehogs injured by cuts and the number of hedgehog care centres reporting more than 5 hedgehogs in several German federal states.

Federal State	Federal State Abbreviation	Number of Reported Hedgehogs	Number of Reporting Care Centres
Baden-Württemberg	BW	42	3
Bavaria	BY	54	2
Berlin	BE	1	0
Brandenburg	BB	8	0
Bremen	HB	5	1
Hamburg	HH	2	0
Hesse	HE	14	1
Mecklenburg-West Pomerania	MV	24	2
Lower Saxony	NI	80	2
Northrhine-Westphalia	NW	107	5
Rhineland Palatinate	RP	0	0
Saarland	SL	5	0
Saxony	SN	12	1
Saxony Anhalt	ST	2	0
Schleswig Holstein	SH	14	0
Thuringia	TH	0	0

**Table 2 animals-14-00057-t002:** Results of the chi-square test for the tested wound parameters with regard to their relationship to overall mortality (euthanasia or died during treatment/care) in 370 reported cut hedgehogs (* *p*-value < 0.05, *** *p*-value < 0.001).

	Mortality Rate [%]	X-Squared	*p*-Value
wound size > 2 × 2 cm	48.9	6.84	0.1447
maggots	51.4	11.73	0.06832
necrosis	50.3	1.67	0.796
injured bones	73.4	2.22	0.6945
abscess	47.1	11.67	0.0199 *
body parts cut off	65.4	11.09	0.02549 *
head	53.8	17.81	0.00013 ***
neck/shoulder	36.6	0.11	0.947
back	38.9	7.02	0.1348
extremities	48.0	1.29	0.5257
flank/abdomen	54.9	60.14	<0.0001 ***
wound age > 12 h	41.3	12.34	0.1947

## Data Availability

All original data (photos and reports) can be found at https://www.facebook.com/groups/igelmitschnittverletzungen/, (accessed on 10 December 2023). Due to graphic nature of the photos and sensitive data (such as the private addresses of gardens in which hedgehogs with cuts were found), this page is only accessible by personal registration with the author.

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
