# Peer review of "Occurrence and Characteristics of Cut Injuries in Hedgehogs in Germany: A Collection of Individual Cases"

_animals, 2023, doi:10.3390/ani14010057_

Round 1

Reviewer 1 Report

Comments and Suggestions for Authors

Introduction

1. The expression "European hedgehogs" in line 39 is incorrect and should read "hedgehogs distributed in Europe", since theoretically there is no subspecies of hedgehog with a European distribution, or the species name of hedgehogs should be standardized throughout the text.

2. What is the role of hedgehogs in German urban ecosystems? What are the consequences of a decrease or increase in their numbers?

3. You should do this study at least on the assumption that you know the distribution range of the hedgehogs distributed in Germany and the approximate population size.

Materials and Methods

1. The data collection period lasted from June 2022 to October 2023, a total of 16 months How did you accurately define whether the hedgehog injuries described by the respondents originated from cuts from the robotic lawnmowers or from other mechanical or human injuries?

2. Is the way of collecting data too limited? You only conducted the survey on facebook, and how many people in Germany do not have internet access? Suggest adding field questionnaires etc. in order to increase the persuasiveness of your data.

3. Are there any statistics on the specific or approximate times when hedgehogs are injured? Define at least day or night.

4. I think it would be useful to demonstrate your point about the following trends in hedgehog injury rates that would be predicted afterwards if mowing was reduced under nocturnal mowing conditions.

5. I think the role of the data uploader is singular and that you don't have statistics on it.

6. I think you should add to the study area map a basic map of the distribution of residential living areas, roads and greenery that has a need for frequent mowing.

7. The reasons for the lack of hedgehog cuts in the RP and TH areas are not discussed and the difference in the number of hedgehogs distributed in the different areas in relation to the frequency of cuts occurring is not discussed, is it that the mowers do not work in these areas or are they less frequent or is there some other reason?

8. I think the simple chi-square test is too homogeneous, and you should add some methods such as model prediction or principal component analysis (PCA) to increase the persuasiveness of your results.

Results

1. If you want to show that grass cutting is an important cause of threats to hedgehog populations, what are the types of hedgehog injuries caused by other anthropogenic disturbances over the same period of time?

2. If you have analyzed the age distribution, sex, etc. of hedgehogs in your results, then you should further assess the impact of the cuts on future population growth trends.

3. What is the significance of your statistics for each day of the week? Or the high incidence of injuries in May-October out of the months of the year is not discussed further.

Author Response

I would like to thank the reviewers for providing relevant and useful comments for my manuscript which I believe has led to the improvement of the paper. I have answered each comment in the attached document.

Reviewer 2 Report

Comments and Suggestions for Authors

The introduction is well structured, describes the biology in the necessary aspects, points out the problems supported by available data and the research question is well described.

Data collection – some parts are redundant and line 115 to 146 my transferred to an appendix. Analysis of the wound may describe in this section.

Figure 1 is quite complicated and contains a lot of information which is not necessary. In my opinion it is enough the show the distribution of the data, also in relation to the care stations. If the authors think, the data in relation to the federal states are important, this should be provided as a separate table.

Line 193-195. A relation is described which is not supported by statistical analysis.

Fig. 5 is quite complicated, especially the legend. I suggest 4 separate graphs e.g. pie chart.

Fig. 6 also mixes up numbers and proportions and is hard to read. Separate graphs will be easier to understand.

Line 244 – not all the readers a familiar with the abbreviations of the german federal states and are not able to make a relation in Fig. 1.

The discussion is well structured to, discusses the findings quit well and puts it into a wider perspective. I would like a comment on the relation between the number of followers of the facebook group and the number of reported hedgehogs.

Author Response

(The authors gave the same response as above.)

Reviewer 3 Report

Comments and Suggestions for Authors

General comments

This is a really interesting and useful piece of work. It must have been quite traumatising going through all the injuries and you are commended for having done that and written this up. Hopefully your hard work will be put to good use and impact upon public policy.

I was surprised by the lack of veterinary input described in the paper. - in fact veterinary care was not really mentioned much at all. I work in wildlife rehabilitation, so I do understand how this can be limited, not least because of a lack of enthusiasm from vets. Veterinary input would have given more technical information about the type of injury and would also have impacted on treatment and prognosis. Do you know if some or all of these animals actually saw a vet? This might affect both how some of the injuries were described and how they were treated, including how rapidly animals were euthanased or released. Clearly what you’ve done is wildlife rehabilitator focused and that’s great, but I think you need to mention the lack or variability of veterinary professional input into these cases. A follow up survey involving vets and veterinary pathologists would be great and might be able to ‘type’ the injuries and shed light on their aetiology.

I think you need to include something more on assessment of the wounds. I assume you did this from photos, or were the wildlife centre staff doing this? How did you reach a conclusion on things like wound age? This needs to be detailed in the methods. If it’s easier to explain with pictures you could selectively include some of those.

I’m not convinced that including historical data from a few centres, back to 2013, before the start of your study, in 2022, is beneficial. Also when you talk about things like differences between the sexes (or outcomes) it would be good to include only those animals where the sex (or outcome) was known - largely because the ‘unknowns’ otherwise confound your data. 

Line 3              I wonder if ‘laceration injuries’ might sound better than ‘cut’? If you do change that (up to you) then do so in other places too e.g. summary and abstract.

Line 19 & 29    ‘died’ or were ‘euthanased’? ‘Died’ suggests natural death. Maybe ‘at least 47% died or were euthanased’. This terminology does get a bit confusing later in the paper. You could use 'did not survive' to include natural death plus euthanasia

Line 25            Is a ‘care centre’ a wildlife rehabilitation centre? If it is it might be better to say that, you use the term early on (line 58)

Line 28            The word ‘only’ is not needed

Line 30            I might say ‘potentially’ or ‘possibly’ rather than ‘presumably’ as we really don’t know if this is the case.

Line 30-33       I might rephrase the sentence starting ‘The comparatively high mortality rate coupled with a presumably ……..’ So it reflects the impact of the situation on both animal welfare and on conservation, something like: The comparatively high mortality rate coupled with a possible high number of unreported cases of hedgehogs with laceration injuries, show that these accidents pose an additional, serious pressure to hedgehogs, impacting both on the welfare of individual animals and having a broader effect on potential conservation of this species.

Line 36            ‘Animal welfare’ would be a better key word than ‘animal suffering’ as it will be picked up in more key word searches

Line 42            maybe ‘habitations’ rather than ‘housings’

Line 54            Assume there is no specific licencing of who can care for hedgehogs in Germany as there is in some other countries (sadly not the UK)?

Line 59-60       You suggest some animals might perhaps not be taken in at all due to costs. Is this not more a ‘triage’ of animals to decide which need to be immediately euthanased and which can be treated, rehabilitated and released? Surely animals ere not just left to die?

Line 62            Unsure what a ‘public’ rehabilitation centre is, is it local authority and funded?

Line 64            Is there a reference for this ‘reporting’ or is it just anecdote?

Line 68-69       I really don’t think you can ‘assume’ at all, it could be due to other things happening concurrently. You should say something like: ‘….it could be hypothesised that the increase in number of hedgehog cut injuries is associated with use of these robotic mowers.’  

Line 80-82       Are there any references for the petitions?

Line 94            maybe ‘graphic nature’ rather than ‘brutality’ of photos

                        Were people asked to record new cases or any cases based on historical records? How many centres actually had decent records? This has a huge impact on Figure 2

Line 97            Maybe better as ‘Every report of a cut injured hedgehog contained the …..

Line 100          Place with a capital P

Line 101          suggest replace ‘cuts’ with ‘injuries’ here

Line 102          You did collect some potential veterinary input info, can this be discussed?

Line 104          Better as something like ‘The following additional information was provided, where known:

Line 107          How could the age of the animal ever be truly ‘known’, wasn’t it always estimated?

Line 112-113   We need to know how the wound characteristics were assessed (using the photos) and by who (yourself I expect). Maybe acknowledging the limitations of this method of assessment (vs a pathological examination), either in methods or discussion, would be good. 

Line 133-140   The quality of the information in these lines will really be affected by who is examining the animal and filling out the sheet. This needs to be acknowledged in the discussion - it is the same issue faced in other studies   

Line 144          ‘spiked armour’ - do you mean spines? You use ‘spines’ in line 146

Line 153          How did you/the rehabilitator know if a wound was >24hrs old? You haven’t said anything about this (assume maggots and necrosis?)? You need to say that this is a presumption. See also line 210. You need something about approximate ‘ageing’ of wounds in here.

Line 154-156   I’m not sure animals with unknown ‘fate’ should be included in the analyses, it would be better to just look at know deaths (did not survive) vs releases - assume all healthy animals were released?

Line 181          This figure is really biased according to if any records had been kept and the quality of that record keeping. I think you need to look at just reports from June 2022 when you started asking for data, or separate out those centres with existing records back to 2013, or whenever, and see how those figures have changed. Line 277-279 describes the more recent data, this might be better data to graph.

Line 210          How do we approximate (and it can only be an approximation) that wounds are ‘older than several weeks’?

Line 222-238   Figure 5 and Table 1 would make more sense if they included only the 295 hedgehogs where a fate was known, otherwise the confidence intervals on these figures are huge and that should be acknowledged. I would personally re do these with the 295 animals.

                        It’s also a bit confusing around the use of the word ‘died’ which is used both for ‘natural deaths’ (not euthanased) and overall mortality (natural death plus euthanasia). Maybe used ‘natural death’ and ‘overall death’ or ‘overall mortality’ or 'did not survive' throughout as appropriate?

Line 228          How can we have a mortality rate that includes animals for which we don’t know the outcome? I think the ‘unknowns’ need to be removed before analysing this data.

Line 251          What’s a ‘hedgehog station’? This is the first time this term has been used. Is it an official rehabilitation centre? Needs explanation or use a different term. 

Line 255          Says the bigger centres did not take part, did this impact on the quality of care overall?

Line 265          Needs a space after the first word and full stop

Line 267          The descriptions maybe ‘suggest that that several or complicated surgeries and expensive medication might be necessary’ - not sure you have the info to say more.

Line 271          Where you say ‘more expensive treatment than other hedgehogs’ unless you have data and references for this, it’s just a presumption. Perhaps ‘are likely to be the more expensive’ might be ok.

Line 283          This is incorrect, hedgehogs do leave the nest during hibernation if the weather is warmer and move around changing nest site. This makes them probably very vulnerable at these times. 

Line 299          Does this fit with robotic mowers perhaps, that are active all the time, not just at weekends? Whereas, as you suggest, people are in gardens at weekends

Line 304          Again it might be useful to consider outcome where fate was known as well

Line 309-310   Not sure if any of the references suggest ‘intensive care patients with an increased 309 probability of mortality’. I’d maybe say something like ‘the variable standards of care at wildlife rehabilitation centres’

Line 315          Again how do we know if animals were found within 24hrs? We need some explanation in the methods, ideally referring to some good references about wound healing. Including who assessed the age of the wound.

Line 316          Unless the hedgehogs told you, this is anecdote, unless you can better grade the age of the wounds.

Line 354          ‘Predators’ better than ‘enemies’

Line 335          This implies the other studies recorded ‘cut injuries’, I don’t think they do, they largely record injured animals and split up those injuries up in various ways. Re read the references and come up with a proper comparison

References      Generally a good use of the available material. Maybe lacking in veterinary references around wound types and healing.

Reference 7     There’s a 2022 version of this reference

Comments on the Quality of English Language

English is generally very good, only a few small comments around rephrasing, which I hope are helpful

Author Response

(The authors gave the same response as above.)

Reviewer 4 Report

Comments and Suggestions for Authors

These authors prepared a manuscript with a detailed analysis of traumatic wounds presented by hedgehogs. From my perspective, this manuscript is well-structured and well-written and gives important information regarding one of the most common reasons for admission of hedgehogs in rescue centres. Therefore, authors should receive credit for their work.

However, I would like to suggest the authors to specifically answer distinct questions and present these answers in their Discussion. The data shown by these authors is more than sufficient to answer these two aspects and I definitely believe these would be very useful for veterinary professionals and hedgehog researchers.

1) Is this a more relevant problem in urban environments or rural environments? Because in urban areas (as public and private gardens) this can be due to the use of personal objects that people use to take care of their gardens. Did you also consider predation wounds? If it is a more rural problem the causes can be different, the machines used in agriculture are also different. I believe this is relevant since more information can be passed to people to use this equipment more carefully through conservation education action. 

2) It would be very relevant to directly compare these reasons for admission/causes of death with others (such as car collisions or debilitation) in order to perceive the magnitude of this problem for hedgehog conservation. I would try to compare your results with other studies that report different percentages of each cause of admission of hedgehogs to rescue centres. At least one I noticed that you already used in your manuscript (Garcês et al.) 

I have nothing further to add.

Author Response

(The authors gave the same response as above.)

Round 2

Reviewer 1 Report

Comments and Suggestions for Authors

It can be seen that the authors have made detailed changes according to the revisions, and most of the experimental logic is more plausible and the data is fuller, and the results can be used to explain the threat that the hedgehog is in Germany.

1.I propose to study the area map and add some elements such as roads, human habitation, etc.

2.Suggestion fig. 2 and 3 to optimize again.

Author Response

See my answers and comments in the attached document.
